# Renin–Angiotensin System Components and Arachidonic Acid Metabolites as Biomarkers of COVID-19

**DOI:** 10.3390/biomedicines11082118

**Published:** 2023-07-27

**Authors:** Biwash Ghimire, Sana Khajeh Pour, Elizabeth Middleton, Robert A. Campbell, Mary A. Nies, Ali Aghazadeh-Habashi

**Affiliations:** 1College of Pharmacy, Idaho State University, Pocatello, ID 83209, USA; biwashghimire@isu.edu (B.G.);; 2Department of Internal Medicine, Division of Pulmonary and Critical Care Medicine, University of Utah, Salt Lake City, UT 84112, USA; 3Department of Internal Medicine, Division ofHematology, University of Utah, Salt Lake City, UT 84112, USA; 4College of Health, School of Nursing, Idaho State University, Pocatello, ID 83209, USA

**Keywords:** renin–angiotensin system, arachidonic acid, COVID-19, biomarkers

## Abstract

Through the ACE2, a main enzyme of the renin–angiotensin system (RAS), SARS-CoV-2 gains access into the cell, resulting in different complications which may extend beyond the RAS and impact the Arachidonic Acid (ArA) pathway. The contribution of the RAS through ArA pathways metabolites in the pathogenesis of COVID-19 is unknown. We investigated whether RAS components and ArA metabolites can be considered biomarkers of COVID-19. We measured the plasma levels of RAS and ArA metabolites using an LC-MS/MS. Results indicate that Ang 1–7 levels were significantly lower, whereas Ang II levels were higher in the COVID-19 patients than in healthy control individuals. The ratio of Ang 1–7/Ang II as an indicator of the RAS classical and protective arms balance was dramatically lower in COVID-19 patients. There was no significant increase in inflammatory 19-HETE and 20-HETE levels. The concentration of EETs was significantly increased in COVID-19 patients, whereas the DHETs concentration was repressed. Their plasma levels were correlated with Ang II concentration in COVID-19 patients. In conclusion, evaluating the RAS and ArA pathway biomarkers could provide helpful information for the early detection of high-risk groups, avoid delayed medical attention, facilitate resource allocation, and improve patient clinical outcomes to prevent long COVID incidence.

## 1. Introduction

COVID-19, a highly contagious disease caused by SARS-CoV-2, is associated with clinical manifestations ranging from asymptomatic to severely diseased states such as pneumonia, acute respiratory distress syndrome (ARDS) and acute lung injury (ALI), multi-organ failure, and even death. COVID-19 has infected over 600 million people during the pandemic resulting in more than 6 million deaths.

The virus enters the cell by binding viral spike protein (S1) to the epithelial cells’ angiotensin-converting enzyme 2 (ACE2), one of the main components of the renin–angiotensin System (RAS) [1]. Recent studies identified soluble ACE2 levels in plasma as a reliable biomarker of COVID-19 [2,3]. Similarly, plasma levels of angiotensin (Ang) peptides and anti-ACE2 auto-antibodies have been considered biomarkers of rheumatoid arthritis [4]. The classical arm of RAS involves ACE/Ang II/Ang II type 1 receptor (AT1R) and controls a diverse range of biological effects through AT1R stimulation by Ang II. On the other hand, the RAS protective arm comprises ACE2/Ang 1–7/Mas receptor (MasR), which counteracts the effects of Ang II [5,6]. Ang II, initially produced by the enzymatic action of ACE on Ang I, plays a pivotal role in regulating blood pressure, electrolyte and fluid balance, and inflammatory pathogenesis. Overstimulation of the classical arm by excessive Ang II produces pro-inflammatory (e.g., ALI), proliferative, and fibrotic effects [7]. ACE2 converts the pro-inflammatory peptide Ang II to the anti-inflammatory peptide Ang 1–7 [8]. Ang 1–7 acts as the RAS’s counter-regulatory peptide, presenting anti-inflammatory, anti-proliferative, and anti-fibrotic effects, and maintaining the body’s hemostasis [9]. The Ang 1–7/Ang II ratio is a critical factor in the activated RAS balance distortion (Figure 1). The binding of the virus to ACE2 leads to its internalization, causing an imbalance between ACE/ACE2, resulting in an unopposed higher level of Ang II and changing Ang 1–7/Ang II, which overactivates the AT1R and by losing the protective effects of the Ang 1–7/Mas R, it starts an inflammatory process that leads to a so-called cytokine storm and initiation and progression of ALI and ARDS [10]. Treatment with ACE inhibitors (ACEi) and Ang II receptor blockers (ARB) theoretically have the potential to prevent and maybe treat ALI and ARDS resulting from COVID-19 infection [11].

Arachidonic Acid (ArA) is released during an inflammatory response by action of phospholipase 2 (PLA2) and then metabolized by different CYP450 enzymes (ω-hydroxylase and epoxygenase) to convert into eicosanoids such as hydroxyeicosatetraenoic acids (HETEs) and epoxyeicosatrienoic acids (EETs), respectively (Figure 2). These eicosanoids function in diverse physiological systems associated with inflammation processes [12,13,14]. For instance, 20-HETE, a potent vasoconstrictor, regulates vascular tone, blood flow to specific organs, and sodium and fluid transport to the kidneys [15]. EETs act as an endothelium-derived hyperpolarizing factor and present with potent vasodilatory and anti-inflammatory effects [16].

Like ACE/ACE2 and Ang II/Ang 1–7 ratios, the balance between 20-HETE/EETs is critical for the body’s physiological homeostasis. Results from adjuvant arthritis animal studies demonstrate that the RAS and ArA pathways are affected, and the balance between their pro and anti-inflammatory axes is disturbed [17,18]. The two pathways are interrelated and directly correlated, demonstrating their significance in the inflammatory response process. Ang II up-regulates phospholipase A, causing ArA release from the cell membrane, propagating the inflammatory cascade furthermore through the ArA pathway [19]. An imbalance between the vasoconstrictor and vasodilator ArA metabolites occurs due to inflammation impacting the RAS and increasing Ang II levels [20]. COVID-19′s inflammatory response affects the RAS and ArA pathways [21]. Given the systemic and local RAS and ArA pathways play essential roles in the homeostasis of vital organs such as lungs, heart, liver, and kidneys, we expect that the understanding and knowledge of the RAS and ArA pathways biomarkers’ status and their association with patient demographic variables (such as gender, age, body mass index (BMI), and underlying comorbidities) could help physicians to predict future complications in these organs and plan for better intervention.

The impact of sex difference is one of the unique traits seen in COVID-19 infections. Recent studies have shown that males are more affected by the disease than females, with higher mortality and hospitalization rate [22,23]. Although the mechanism for this discrepancy has not been elucidated, recent communication reports attributed it to an infectivity mechanism of COVID-19 involving the ACE2 receptor, Type II transmembrane Serine Protease (TMPRSS 2) and androgen receptor [24]. TMPRSS 2 is vital in activating viral spike protein, a crucial step in viral entry. The androgen receptor is a known promoter of TMPRSS2 gene expression [25]. The upregulation of this gene by the androgenic stimulation could explain the higher infectivity in males [26]. In addition, lesser infectivity in children could be explained by the lower expression of the androgen receptors due to underdeveloped sex organs. The phenomenon has been reported for sex hormones’ influence on ACE2 expression and activity in the mouse adipose tissue, kidneys, and myocardium [27,28,29]. While speculation, it seems reasonable that if the sex hormone modulator 17β-Oestradiol affects the expression and activity of the ACE2 receptor and increases Ang 1–7 levels; thus, hormone therapy offers a potential supportive treatment for female COVID-19 patients [30].

Based on the data provided by the U.S. Centers for Disease Control (CDC), rates of hospitalization and death increase with age. Compared to the age group of 18–29 years, the hospitalization rate was 5× higher in the 65–74 and 9.1× more in the 75–84 age groups. Similarly, the mortality rate increased by 25× in the 50–64, 65× in the 65–74 and 140× in the 75–84 age groups [31]. COVID-19 hospitalizations rate could also be correlated with obesity and BMI. A study from Johns Hopkins showed that the hospitalization rate increased in adults with obesity and underweight relative to the normal weight [32]. Obesity is also linked to impaired immune function and reduction in lung capacity, thereby increasing the risk of infection and impairing ventilation. As reported by CDC, increased BMI could be related to an increased risk of ICU admission, invasive mechanical ventilation, and death [33].

This study aims to analyze the effector RAS peptides (Ang II and Ang 1–7) and relate them with the metabolites of the CYP450-mediated ArA pathway. Additionally, we correlated these metabolites with the demographic variables known to worsen the prognosis of COVID-19 to understand how these variables play a role at the molecular level in the infection. The results of this study help explain how variability in biomarkers combined with individuals’ demographics and comorbidities correlate with COVID-19 disease intensity. Such knowledge could allow healthcare providers to predict future complications and plan for better interventions.

## 2. Materials and Methods

### 2.1. Materials

Ang 1–7 (Anaspec, AS-61039) and Ang II (Anaspec, AS-20633) were acquired from Anaspec Inc. (Fremont, CA, USA). (Asn1, Val5)-Ang II (IS), Sigma-Aldrich A6402-1MG, was obtained from Sigma Aldrich (St. Louis, MO, USA). Waters C18 SPE cartridges (Sep-Pak WAT020805) were purchased from Waters (Milford, MA, USA). The reference standards for ArA metabolites were procured from Cayman Chemical Company (Ann Arbor, MI, USA). These included 19-(R)-hydroxyeicosatetraenoic acid (19-HETE) (P/N,10007767), 20-hydroxyeicosatetraenoic acid (20-HETE) (P/N,90030), (±)-5,6-epoxyeicosatrienoic acid (5,6-EET) (P/N,50211), (±)8,9-epoxyeicosatrienoic acid (8,9-EET) (P/N,50351), (±)11,12-epoxyeicosatrienoic acid (11,12-EET) (P/N,50511), (±)14,15-epoxyeicosatrienoic Acid (14,15-EET) (P/N,50651), (±)-5,6-dihydroxyeicosatrienoic acid (5,6-DHET) (P/N, 51211), (±)8,9-dihydroxyeicosatrienoic acid (8,9-DiHT) (P/N,51351), (±)11,12-dihydroxyeicosatrienoic acid (11,12-DHET) (P/N,51511), and (±)14,15-dihydroxyeicosatrienoic acid (14,15-DHET) (P/N,51651). The deuterated internal standards (IS) listed below were also obtained from Cayman, with the deuterium atoms present at the carbons 16, 17, 18, 19, and 20 positions and an isotopic purity of ≥99% for 8,9-EET-d11. LC-MS grade water, acetonitrile, and formic acid were purchased from Fisher Scientific (Fair Lawn, NJ, USA).

### 2.2. LC-MS/MS System

LC-MS/MS system comprised of liquid chromatography (Shimadzu, MD, USA) with a binary pump (LC-30AD), an autosampler (SIL-30AC), a controller (CBM-20A), a degasser (DGU-20A5R), a column oven (CTO-20A), and an ABSciex QTRAP 5500 mass spectrometer (SCIEX, Foster City, CA, USA) with electron spray ionization (ESI) source. The chromatograms were monitored using Analyst 1.7 software, and the data were analyzed in MultiQuant 3.0 software (SCIEX, Foster City, CA, USA). The analytes were separated using Synergi™ Fusion-RP column (2.5 µm, 100 × 2 mm) obtained from Phenomenex (Torrance, CA, USA). All the analyses were performed in positive ion mode.

### 2.3. Human Subject

The Institutional Review Board approved this study through the IRB-FY2020-273 protocol to analyze de-identified COVID-19 patients’ plasma samples provided by the University of Utah Hospital under a signed Material Transfer Agreement. These patients were recruited for another study at the University of Utah which the institution’s IRB also reviewed and approved all study recruitment materials, educational information, participant instructions for self-collection of specimens, surveys, and the informed consent documents (IRB-00102638) and (IRB-00093575).

The University of Utah research group inclusion criteria for healthy volunteers were consenting male and female subjects of any self-identified race/ethnicity without acute or chronic illnesses aged 18 and older, and exclusion criteria were the use of any prescription medication, pregnancy, suffering from any acute or chronic medical condition or disease, having been hospitalized or had surgery within the preceding 6 weeks, or having any prior history of stroke. The inclusion criteria for COVID patients were ICU admission with confirmed infection and organ dysfunction as defined by a Sequential Organ Failure Assessment (SOFA) ≥ 2 above baseline (if baseline data are unavailable, baseline SOFA is assumed to be 0). All patients were enrolled in the study within 72 h of ICU admission. Additionally, individuals diagnosed with COVID and hospitalized with acute COVID infection were enrolled in the study. The exclusion criteria were admission to the ICU for longer than 72 h, hemoglobin level < 7 gm/dL, or clinically significant bleeding.

After the patient met inclusion criteria and no exclusion criteria were indicated, subjects provided consent using approved IRB protocols. Once subjects consented, a whole blood sample was drawn within 48 (±24) h (study day 0) of a COVID-19 diagnosis. Aliquots of blood samples were mixed with a protease inhibitor cocktail solution containing 1.0 mM p-hydroxy mercury benzoate, 30 mM 1,10-phenanthroline, 1.0 mM phenylmethylsulphonyl fluoride, 1.0 mM Pepstatin A, and 7.5% EDTA (all from Sigma Aldrich, St. Louis, MO, USA), and treated with 1% Triton X (to inactivate the virus). Plasma was harvested after centrifugation for 10 min at 2500× *g* and 4 °C and stored at −80 °C until assayed.

### 2.4. Sample Preparation Procedures

#### 2.4.1. Solid-Phase Extraction (SPE)

Angiotensin peptides (Ang 1–7 and Ang II) were extracted from the plasma samples using solid phase extraction (SPE). SPE was carried out based on a previously established method by Cui et al. with minor changes in the process [34]. Briefly, 200 μL of plasma samples was mixed with 100 μL of the internal standard (50 ng/mL). Then, 1.5 μL of formic Acid was added to make the final concentration of 0.5%. The samples were mixed and loaded onto the Waters C18 SPE cartridges previously preconditioned with 3 mL ethanol and 3 mL deionized water. Samples were then loaded onto the cartridge and allowed to interact with the column by application of a positive nitrogen flow from a positive pressure manifold (Agilent Technologies Inc., Santa Clara, CA, USA). It was then washed with 3 mL of deionized water and eluted with 3 mL of methanol containing 5% formic acid. The eluted solutions were collected and dried using a Savant 200 SpeedVac system (Thermo Fisher Scientific, Waltham, MA, USA). The dried samples were reconstituted in 100 µL of acidified water containing 0.1% formic acid, and 10 µL of samples were injected into the LC-MS/MS to quantify the Ang peptides concentrations.

#### 2.4.2. Liquid–Liquid Extraction (LLE)

A liquid–liquid extraction (LLE) method using ethyl acetate was utilized to extract the ArA metabolites from the plasma samples. This method was previously validated in our lab [35]. Briefly, 300 µL of plasma sample was mixed with 100 µL of 10,11-EET d11 (IS, 100 ng/mL), and 2 µL of FA was added. The samples were vortexed, and 500 µL of ethyl acetate was added. The resulting biphasic solutions were mixed thoroughly by vortexing for 1 min and centrifuged at 15,000× *g*, 4 °C, for 15 min. After centrifugation, 400 µL of the supernatant layer was taken, and for a second extraction step, 500 µL of the ethyl acetate was added again. The organic supernatant phase was mixed with the previous step extract, dried under nitrogen gas, and reconstituted in methanol for LC-MS/MS analysis. A volume of 30 µL of the sample was injected into the LC-MS/MS.

#### 2.4.3. LC-MS/MS Method for Ang Peptides

Ang 1–7 and Ang II levels in the patients’ plasma samples were analyzed using the LC-MS/MS method described in a published study [34]. Minor modifications and validation were performed before applying the analytical method for plasma sample analysis. Separation of Ang peptides was achieved using a Synergi RP column (2 × 100 mm) with a particle size of 2.5 μm from Phenomenex (Torrance, CA, USA). The column was maintained at ambient temperature. The mobile phase contained water with 0.1% formic acid (A) and ACN (B).

The gradient time program was initiated with 5% ACN and increased to 30% ACN over a 4-min period. The composition was held steady from 4 to 8 min, then decreased to 5% ACN at 9 min and maintained for 10 min. The flow rate was set at 0.3 mL/min, and the injection volume was 30 μL. Electrospray ionization was employed, and the analytes were detected using multiple reaction monitoring (MRM) in positive mode. The optimized parameters for the source/gas were as follows: curtain gas, 30; collision gas, medium; ion spray voltage, 5500 V; temperature, 300 °C; ion source gas 1 (nebulizer gas), 20 psi; and ion source gas 2 (turbo gas), 25 psi.

LC-MS analysis was conducted using the single ion recording (SIR) mode, with specific *m*/*z* values: 300.5 for Ang 1–7, 349.6 for Ang II, and 516.6 for the internal standard (IS). For LC-MS/MS, the MRM transitions used were *m*/*z* 300.6→136 for Ang 1–7, *m*/*z* 349.6→136 for Ang II, and *m*/*z* 516→769.4 for the internal standard.

#### 2.4.4. LC-MS/MS Method for ArA Metabolites

The experimental protocol and assay conditions were conducted with minor optimization to analyze ArA metabolites, following a previously described method [35]. In summary, these metabolites were separated using a Synergi^TM^ RP column (2 × 100 mm) with a particle size of 2.5 μm from Phenomenex (Torrance, CA, USA). The separation was carried out at ambient temperature. The mobile phase contained water with 0.1% formic acid (A) and ACN (B).

The gradient time program for the mobile phase began with 5% ACN and increased to 20% ACN within 2 min. It was then raised to 55% ACN and maintained for 2.5 to 6 min before rising to 100% ACN at 8 min. The method continued until 9 min, after which the mobile phase was reduced to 5% ACN over a total run time of 10.5 min. The flow rate was set at 0.3 mL/min, and the injection volume was 10 μL.

For mass spectrometric analysis, a triple quadrupole was utilized to monitor the *m*/*z* transitions using the Analyst^®^ 1.7.2 software. Electrospray ionization was employed, and the analytes were detected using multiple reaction monitoring (MRM) in the negative mode. The optimized parameters for the source/gas were as follows: curtain gas, 20 psi; collision gas, medium; ion spray voltage, approximately 4500 V; temperature, 400 °C; ion source gas 1 (nebulizer gas), 20 psi; and ion source gas 2 (turbo gas), 25 psi.

### 2.5. Statistical Analysis

Patients’ demographic data are presented as mean ± standard deviation (SD). The biomarkers data are expressed as mean ± standard error of means (SEM) and analyzed by a standard computer program, GraphPad Prism Software PC software, version 9.3.1, and Statistical Package for Social Sciences (SPSS) version 28 for Windows (SPSS Inc., Chicago, IL, USA). The plasma concentrations of some of the ArA metabolites in some patients were lower than the LC-MS/MS method detection limit. In addition, the level of Ang II (in three cases) fell outside the 3 standard deviations threshold based on the Z-score method and were considered outliers. Therefore, those patients were excluded in the statistical comparison case by case. Data were tested for normal distribution using the Kolmogorov–Smirnov test and homogeneity of variance using Levene’s test before proceeding with the non-parametric statistical tests. Group comparisons were made using the Mann–Whitney U test, and the correlation for the continuous variables was analyzed using Pearson’s correlation coefficient. The confidence interval was set at 95%, and *p* < 0.05 was considered statistically significant.

## 3. Results

### 3.1. Characteristics of Healthy and COVID-19 Infected Cohort

This study analyzed de-identified plasma samples from 6 healthy subjects and 27 COVID-19 patients. The demographic variables assessed were gender, age, and obesity, and the medication use, as well as a diagnosis of diabetes, hypertension, respiratory comorbidities, and SOFA score, were considered (Table 1).

The gender composition of the control group was 33% male, and COVID-19 patients were 57.14% male. The higher percentage of males in the patient group can indicate that males are more affected than females; however, in this study, it was only a slight difference between the numbers of male and female participants. The average age for the control group was 34.67 ± 16.23 years, with one participant above the age of 65. Nine of the twenty-seven (33.3%) patients were older or equal to 65 years old. Most participants (66.7%) in this study were younger than 65. The subjects BMI distribution varied from underweight (BMI < 18.5), including 1 (3.70%) participant; normal (BMI = 18.5–24.9), including 4 (14.81%) participants; overweight (BMI = 25–29.9), including 10 (37.04%) participants; obese (BMI = 30–34.9), including 5 (18.52%) participants; and highly obese (BMI > 35), including 8 (29.63%) participants. Participants were categorized into two categories based on their BMI being less than or more than 30. 

Participants reported cardiovascular 22 (81.48%) and respiratory 10 (37.04%) comorbidities. The major cardiovascular comorbidities cases were diabetes 15 (55.56%), hypertension 13 (48.15%), and cardiac arrhythmia 4 (14.81%). Regarding respiratory disease, six (22.22%) participants had asthma, two (7.41%) had COPD, and two (7.41%) suffered from other lung diseases. There were 10 (37.04%) current or post-smoker patients. Participants’ medical use history indicates that they were taking statins (10, 37.04%), nonsteroidal anti-inflammatory drugs (NSAIDs) (5, 18.52%), Aspirin (4, 14.81%), and corticosteroids (2, 7.41%). The SOFA score of the participants was 5.29 ± 2.91, while their platelets count, total bilirubin, serum creatinine, and hemoglobin levels were in the normal range. The average WBC (per μL) (10.45 ± 5.52 × 10^3^) was above the normal values (Table 1).

### 3.2. Effect of COVID-19 Infection on the RAS Components Levels

The plasma levels of Ang 1–7 and Ang II and their ratio in healthy individuals and COVID-19 patients are presented in Figure 3 and Table 2. The Ang 1–7 level was significantly (*p* < 0.0001) reduced in the patients infected with COVID-19 (1.37 ± 0.20 ng/mL) compared to the control (23.00 ± 5.69 ng/mL) group (Figure 3A). The Ang II concentration, on the other hand, was significantly (*p* = 0.0002) elevated in the COVID-19 patients (4.61 ± 0.92 ng/mL) compared to the control group (0.29 ± 0.09 ng/mL) (Figure 3B). The Ang 1–7/II ratio as an indicator of RAS imbalance was also significantly (*p* < 0.0001) lower in COVID-19 patients (0.96 ± 0.26) than in healthy control individuals (85.81 ± 17.90) (Figure 3C).

### 3.3. Effect of COVID-19 Infection on ArA Metabolites

The CYP450-mediated ArA pathway metabolites profiles in healthy control and COVID-19 patients are presented in Table 2. This study measured ten metabolic products of this pathway (two HETEs, four EETs, and four DHETs metabolites). COVID-19 patients did not present with a significant increase in 19-HETE and 20-HETE pro-inflammatory metabolites, but there was a considerable change in anti-inflammatory EETs and DHETs metabolites’ profiles. The level of EETs was dramatically increased in COVID-19 patients, whereas the DHETs concentration was repressed. The total-EETs were significantly (*p* < 0.0001) elevated in the patients’ group (71.05 ± 12.64 ng/mL) compared to the control group (9.83 ± 0.84 ng/mL). The most prevalent EET in the patients’ group was 11,12-EET (41.93 ± 7.79 ng/mL), which was a significantly higher (*p* < 0.0001) than that of the control group (1.96 ± 0.47 ng/mL). Similarly, concentrations of 8,9-EET (*p* < 0.05) and 14,15-EET (*p* < 0.01) were significantly increased in the patients, whereas the 5,6-EET level was comparable between groups (*p* > 0.05). The concentration of all four DHETs, as metabolites of EETs, was suppressed due to the COVID-19 infection and were significantly lower in the patients’ group than in the control group (Table 2).

### 3.4. Correlation of the RAS Components and ArA Pathway Metabolites

The correlation of RAS metabolites level with the ArA metabolites was tested to explain how dysregulated RAS affects the ArA pathway status (Table 3, Table 4 and Table 5). In the healthy control group, there was no significant correlation between RAS components and ArA metabolites other than in a few cases. The total-EETs level positively correlated Ang 1–7 (r = 0.8559, *p* = 0.0296) (Table 3). A similar positive correlation was observed between Ang II with 5,6-DHET (r = 0.8401, *p* = 0.0363) (Table 4) and Ang 1–7/Ang II with 8,9-EET (r = 0.8589, *p* = 0.0285) (Table 5). In the case of the COVID-19 patients, the Ang 1–7 levels were negatively correlated with 20-HETE (r = −0.4089, *p* = 0.0489), while it was not associated with other metabolites. On the other hand, the Ang II level was positively correlated with 19-HETE (r = 0.5052, *p* = 0.0195), total-HETE (r = 0.5317, *p* = 0.0131), 5,6-EET (r = 0.6805, *p* = 0.0007), 14,15-EET (r = 0.6721, *p* = 0.0008), total-EEts (r = 0.5998, *p* = 0.0041), and 8,9-DHET (r = 0.6290, *p* = 0.0068). In the case of the Ang 1–7/ang II ratio, there was a trend of negative correlations with ArA metabolites; however, it did not reach a significant level.

Regardless of their disease status, all participants’ Ang 1–7 level was negatively correlated with 11,12-EET (r = −0.3899, *p* = 0.0597) and total-EET (r = −0.3874, *p* = 0.0557). It was positively correlated with 5,6-DHET (r = 0.7496, *p* = 0.0002), 11,12-DHET (r = 0.6704, *p* = 0.0122), and 14,15-DHET (r = 0.5930, *p* = 0.0018). Although there was a positive correlation trend between Ang 1–7 and 8,9-DHET (r = 0.4141), it did not reach a significant level (*p* = 0.0695). Interestingly, the Ang II level did not correlate significantly with any ArA metabolites. The ratio of Ang 1–7/Ang II was strongly correlated with all DHETs and followed a similar trend as of Ang 1–7 (Table 6).

### 3.5. Effect of Biological Variables on the RAS Components and ArA Metabolites Level

Based on the age of the patients, they were divided into geriatric (≥65) and non-geriatric (<65). Ang 1–7, Ang II levels and their ratio were not affected by the age of the patients (Figure 4, upper panel). Similarly, the Ang 1–7, Ang II levels and the ratio of Ang 1–7/Ang in male patients were comparable to female patients. Body weight differences in COVID-19 patients did not impact the Ang 1–7 and Ang II levels. A decrease in Ang 1–7/Ang II ratio in patients with BMI > 30 was insignificant (Figure 4, lower panel).

The COVID-19 patients’ age, gender, and BMI impacts on ArA metabolites follow similar patterns to the RAS components, which did not reach statistically significant differences.

### 3.6. Effect of Comorbidities on the RAS and ArA Pathway

The impact of existing comorbidities (respiratory, cardiovascular, hypertension, or diabetes) on the RAS components and ArA metabolites was tested in COVID-19 patients. Our data indicate no significant differences in the level of the RAS components in patients with and without comorbidity (Figure 5). Similarly, in the case of ArA metabolites, the effect of the presence or absence of comorbidities was insignificant.

Finally, the correlation of the Sequential Organ Failure Assessment (SOFA) score, a scoring system used to assess the performance of several organ systems in the body, was tested in COVID-19 patients. The higher the SOFA score, the higher the likely mortality. The SOFA score was positively correlated with Ang II (r = 0.41717, *p* = 0.0426). The sign of such correlation was negative with the 19-HETE (r = −0.4842, *p* = 0.0165) and total-HETE (r = −0.4828, *p* = 0.0169). The EET levels did not present any correlations (Table 7).

There was no significant correlation between other clinical characteristics (platelet count, total bilirubin, serum creatinine, WBC, and hemoglobin) with the RAS components and ArA metabolites level.

## 4. Discussion

In this study, we evaluated and compared the RAS components (Ang 1–7, Ang II plasma levels, Ang 1–7/Ang II ratio) and ArA metabolites (HETEs, EETs and DHETs) between healthy controls and COVID-19 patients to determine whether the RAS axes were unbalanced in COVID-19 patients and impacted on the ArA pathway and its metabolites profiles.

Our results indicate that Ang 1–7 levels were significantly lower, whereas Ang II levels were prominently higher in the COVID-19 patients than in the healthy control individuals (Figure 3 and Table 2). The ratio of Ang 1–7/Ang II as an indicator of the balance between the RAS classical and protective arms was dramatically lower in COVID-19 patients. Such imbalance is attributed to the deactivation of the ACE2 enzyme by the SARS-CoV-2 virus and matched with measured higher Ang II plasma levels in these patients. Our study’s results agree with most of the previous reports that showed that the protective arm of the RAS is downregulated due to the infection [36,37,38,39]. In concert with our findings, these studies found significantly lower circulating Ang 1–7 levels in COVID-19 patients than in the control group. They also reported that the Ang II levels were lower in the patients that recovered from the disease than those who succumbed to the infection, and, similarly, concluded that the elevated ratio of Ang II/Ang 1–7 is an indication that the balance of the RAS shifted towards the classical axis with increasing hospitalization, disease severity, and mortality [36]. In another study, Lieu et al. found plasma Ang II levels to be elevated in the COVID-19-infected groups compared to healthy controls [38]. Additionally, the study of Wu et al. determined that the circulating plasma concentration of Ang II increased with the severity of the disease [39].

Although our study results align with most recently reported literature [36,37,38,39], few studies have reported that Ang 1–7 levels were elevated in the infected group rather than in the controls [40,41,42]. The observed controversy could be explained by different methodological approaches in patient enrolment or sample collection and analysis, among other possible explanations. Martins et al. analyzed plasma samples from individuals already taking ACEi or ARBs [40]. These classes of drugs are known to increase ACE2 activity, elevate Ang 1–7, and reduce Ang II plasma levels. Such an effect is expected to be more pronounced in healthy controls as no virus is present to inhibit ACE2 enzyme activity, somehow explaining the observations of Martins et al. [40]. A comprehensive meta-analysis showed that the levels of Ang 1–7 in some studies were approximately 11-fold higher in the infected group than in the control group. It also included other controversial studies that reported the other way around. Interestingly, this meta-analysis founds that the level of Ang II was also elevated in the COVID-19-infected group [40]. The observed contradiction could have occurred due to including a range of studies that utilized different sample collection and processing methods. Some of those studies did not consider the enzymes and peptide stability, and some used nonspecific analytical methods, as highlighted by Martins et al. [40]. In another small study (enrolling 10 patients and 5 healthy control subjects), authors looked at equilibrium analysis reflecting overall plasmatic RAS activity instead of its actual status as they did not consider adding protease inhibitors to plasma samples [41]. That means the reported results are not representing the actual status of the RAS components at the time of sample collection. Therefore, it is not a direct measure of RAS peptides, but measures the ongoing activity of RAS proteases which could make a significant difference in the level of peptides. It is worth mentioning that the RAS is a complex system comprised of non-ACE pathways that could impact the Ang II and Ang1–7 levels when ACE2 activity was influenced by COVID-19 infection [41].

It is worth mentioning the observed results indicate that both arms of the RAS are affected by this viral infection. The binding of the virus to ACE2 leads to its down-regulation, causing an imbalance between ACE/ACE2 and consequently resulting in a lower Ang 1–7/Ang II ratio and a surge of an unopposed Ang II and cytokine storm to impose tissue injury depicted in Figure 1, signifying the lung as a typically impacted primary tissue by the virus. The RAS is closely linked to cardiovascular function, and its dysregulation in COVID-19 can have cardiovascular implications. Ang II can promote vasoconstriction, oxidative stress, and prothrombotic effects, increasing the risk of cardiovascular complications reported in infected individuals. The inflammatory response triggered by the dysregulated RAS can damage endothelial cells, disrupt the integrity of blood vessels, and contribute to cardiovascular dysfunction. Such phenomenon has been supported by an animal model of arthritis, where an imbalance of the cardiac and renal RAS components explains the cardio-renal toxicity [17]. Treatment with ACE inhibitors (ACEi) and Ang II receptor blockers (ARB), which are commonly used to manage hypertension, could be beneficial in COVID-19 by potentially mitigating the harmful effects of an imbalanced RAS [42]. However, more research is needed to fully understand the complex interactions between RAS and COVID-19 and determine the optimal therapeutic approaches.

We also observed that in COVID-19 patients, there was no significant increase in inflammatory 19-HETE and 20-HETE; however, there was a remarkable change in anti-inflammatory EETs and DHETs profiles (Table 2). The level of EETs was dramatically increased in COVID-19 patients, whereas the DHETs concentration was repressed. ArA metabolism is a complex process involving several enzymes, including CYP450 enzymes, which play a role in converting ArA into various bioactive metabolites, such as HETEs, EETs, and DHETs. Research on the specific impacts of COVID-19 on ArA metabolites through CYP enzymes is limited; however, it is known that COVID-19 can lead to a dysregulated immune response and excessive inflammation in some individuals. Inflammatory processes, including the production of ArA metabolites, are tightly regulated by various factors, such as cytokines and other inflammatory mediators. It is plausible that COVID-19-induced inflammation could indirectly impact the ArA pathway by altering the expression or activity of enzymes involved in its metabolism, including CYP enzymes. It has been reported that CYP enzyme expression was affected by the excessive inflammatory response due to COVID-19 viral infection [43]. For example, CYP4A1 and CYP4A2 enzymes convert ArA to HETEs and promote the expression of inflammatory cytokines and adhesion molecules [44].

Furthermore, the CYP epoxygenase enzyme families of CYP2C and CYP2J generate EETs from ArA, resulting in anti-inflammation, vasodilation, and proangiogenic effects [45]. Multiple studies demonstrated that both EETs and HETEs play a role in lung and kidney injury [46,47,48]. These reports suggested that the dysregulation of these enzymes may contribute to the inflammatory response observed in COVID-19 patients. In the current study, the observed increase in EETs and decrease in DHETs levels could be attributed to the body’s defense mechanism in resolving the inflammation by up-regulation of CYP epoxygenase enzymes and down-regulation of the sEH. It is possible that other enzymatic mechanisms could also be responsible for this phenomenon, and the proposed claim needs further investigation to be confirmed.

Evidence elucidates that the cross-talk between the RAS and CYP-mediated ArA pathway can significantly affect inflammatory disease manifestations [18] and explain vascular complications observed in COVID-19 patients. A possible link between ArA metabolites level and ACE enzyme induction has been reported [49]. Consequently, the elevated 20-HETE level and low EETs plasma concentrations in patients associated with renal and vascular complications were correlated with plasma renin activity [50]. Additionally, investigation of the association between 20-HETE and the RAS components in rats has shown similar patterns on increased blood pressure and over-expression of CYP4A2 cDNA, which was normalized by the administration of lisinopril, losartan, or a 20-HETE antagonist [51]. The different results on the ArA metabolites observed in this study could be explained by the activation of AT2R by the high concentration of Ang II in COVID-19 patients. It has been reported that AT2R activation favors EETs production, inhibits pro-inflammatory cytokine intracellular signaling [52], and probably helps patients to recover from this viral infection. The anti-inflammatory effects of direct AT2R stimulation have been reported using selective peptide [35] and nonpeptide [53] AT2R agonists. These anti-inflammatory effects do not counteract Ang II-induced, AT1R-mediated pro-inflammatory actions. AT2R- stimulation antagonizes the effects of TNF-α or other non-RAS stimuli, such as growth factors and has been reported previously [54,55,56]. The Ang II signaling through AT2R activation seems to interfere with other non-RAS signaling cascades coupled with harmful stimuli, and ARBs seem unable to block cytokine-induced IL-6 expression [57]. AT2R stimulation reduces TNF-α–induced IL-6 expression by activating protein phosphatases, increasing EETs synthesis and inhibiting NF-κB activity [52]. The observed higher concentration of Ang II in COVID-19 patients due to internalization and inactivation of the ACE 2 could explain the elevated EETs levels and their positive correlation with Ang II levels (Table 2, Table 3, Table 4, Table 5 and Table 6). Previous reports on the protective effects of AT2R in other inflammatory conditions support this finding [58,59].

Additionally, it has been reported that in an in vitro study, Ang 1–7, through activation of the Mas receptor, increased the release of ArA, which the Mas antagonist abolished. Neither AT1R nor AT2R antagonists could block this effect [60]. The current study’s results indicate a negative trend of correlations between EETs and Ang 1–7 or Ang 1–7/Ang II in COVID-19 patients is in concert with that report (Table 3 and Table 5).

Arachidonic acid is a polyunsaturated fatty acid generated by membrane phospholipids under inflammatory conditions facilitated by PLA2. It has been proposed that ArA possesses antiviral properties and can contribute to the inactivation of enveloped viruses such as SARS-CoV-2 [61]. ArA can serve as a substrate for various pathways, including cyclooxygenase (COX), lipoxygenase (LOX), and CYP-mediated epoxygenation and hydroxylation pathways, leading to the production of diverse metabolites. These metabolites play a role in regulating inflammation, immune responses, and vascular function. The dysregulation of ArA metabolites has been implicated in multiple diseases, including respiratory disorders. The COX pathway generates pro-inflammatory lipid mediators such as prostaglandins (PGs) and thromboxane (TXA2). The LOX pathway, on the other hand, produces leukotrienes and lipoxins, which exhibit both pro- and anti-inflammatory activities. In contrast to the pro-inflammatory lipid mediators, certain ArA mediators, such as lipoxins, contribute to inflammation resolution and possess potent anti-inflammatory effects [61]. Among the various PGs, PGE2 has been suggested to play a significant role in COVID-19 pathophysiology by promoting hyperinflammatory and immune responses [62].

Epoxyeicosatrienoic acids (EETs) and their metabolite, dihydroxyeicosatrienoic acid (DHET), exert various biological effects on the vascular system and interact with other ArA metabolites (e.g., prostanoids) and pathways such as the renin–angiotensin system (RAS). A recent study reported that EETs and prostanoids interact at the receptor level, with EETs antagonizing the actions of vasoconstrictor lipids [63]. Therefore, the higher levels of EETs, as observed in our study, could be considered the body’s defense mechanism in the fight against COVID-19.

Regarding gender differences, it has been observed that the expression of COX-2 and PGE2 increased more in males compared to females during acute inflammation in humans [64]. Based on this finding, elevated PGE2 levels in males may contribute to more severe COVID-19 infections in this group [62]. Considering that COVID-19 patients with comorbidities, including obesity, face a higher risk of disease aggravation, PGE2 could also contribute to the severity of the disease in these patients, as increased PGE2 levels have been reported in obese individuals [65]. Although the results of this study showed differences in the RAS component and ArA metabolites levels based on age, sex, BMI, and comorbidities, such distinctions did not reach a significant level (Figure 4 and Figure 5). The lack of detection of such differences could be related to our study’s limitation on enrolling a small number of patients. It is most likely that such a distinction could be made in more extensive studies.

It has been reported that the SOFA score can be used as a tool in combination with other disease severity scores in predicting mortality in COVID-19 patients [66]. Our observed association can explain the potential and positive correlation of the SOFA score with Ang II, 19- and total-HETE, indicating that these RAS components and ArA metabolites directly or indirectly impact COVID-19 patients’ organ function.

The evolving landscape of breakthrough infections, variants of concern, and vaccination can have potential implications for the efficacy of the components of the RAS and ArA metabolites as identified biomarkers. Addressing these considerations can enhance the practical impact of the associations between the RAS, ArA metabolites, and these infections. This goal can be achieved by considering the specific variants of concern and their effects on these biomarkers through more extensive studies to establish the associations that can be better contextualized in the evolving landscape. The validation of our findings and understanding of how these biomarkers relate to specific variants or breakthrough cases can help identify individuals at higher risk for severe infection or who may respond differently to particular treatments. This knowledge enhances risk stratification, treatment guidance, and long-term health management strategies, enabling more personalized and practical approaches in the context of the evolving landscape of these infections and vaccination efforts.

Based on the current study’s findings, it may be reasonable to consider the potential use of NSAIDs in the early treatment of COVID-19 when there are no apparent contraindications. However, this issue has been a subject of considerable debate and research [67,68,69]. One factor that sparked interest in using NSAIDs is the imbalance of the RAS in favor of the proinflammatory peptide Ang II in COVID-19 patients, and these anti-inflammatory agents could restore it, as was shown in animal and human studies [17,70,71]. Ang II promotes inflammation and vasoconstriction, and its increased levels have been associated with severe lung injury and acute respiratory distress syndrome in COVID-19 patients. Early treatment with NSAIDs aims to modulate the immune response [72]. It may impact the pro-inflammatory mediators such as Ang II, ArA metabolites, and cytokine storm triggered by the viral infection, potentially preventing disease progression and the need for hospitalization [18,73]. Although ample evidence supports such beneficial effects of NSAIDs on the management of inflammatory conditions, studies have shown that using these agents may have negative or no outcome at all [68,69]. However, using NSAIDs with or without other anti-inflammatory drugs may be beneficial in reducing hospitalization and mortality if intervened in the early stage of the viral infection [74,75]. To conclusively elucidate the effect of NSAIDs, further extensive clinical trials are required to confirm the same findings in managing COVID-19.

### Limitations

One significant limitation of this pilot study was the small number of patients, limiting its potential for generalizability. This study’s findings and conclusions may not apply to a larger population or diverse demographics, as the sample size may not adequately represent the variations and complexities present in the broader population. However, increasing the study’s statistical power by enrolling a larger population could enhance the ability to detect significant correlations accurately, as some trends have already been seen.

## 5. Conclusions

Despite the limitations associated with the small sample size of this study, it serves as a valuable starting point for generating hypotheses and exploring preliminary trends. We identified potential associations or patterns that can later be investigated in larger, more rigorous studies. The established association of the RAS in the pathogenesis of COVID-19 and the involvement of other cross-talking pathways, such as the ArA pathway, in disease progress and their impact on the patient’s outcome highlights their importance and suggest the usefulness of assessment of their components level as biomarkers of COVID-19 for diagnostic and prognostic tools. Our previous study [4] showed that plasma angiotensin peptide levels are reliable biomarkers of an inflammatory condition such as rheumatoid arthritis. This could be considered a gold standard when applied to COVID-19 as another inflammatory condition associated with the imbalanced RAS and ArA pathway.

Our findings demonstrate notable differences between COVID-19 patients and healthy controls regarding Ang 1–7 and Ang II levels. The Ang 1–7 levels were significantly reduced, whereas Ang II levels were elevated in COVID-19 patients compared to healthy individuals. Furthermore, the ratio of Ang 1–7/Ang II, which serves as an indicator of the balance between the classical and protective arms of the RAS, was considerably lower in COVID-19 patients. The study did not reveal a significant increase in inflammatory ArA metabolite markers. However, the concentration of EETs exhibited a substantial increase in COVID-19 patients, while the concentration of DHETs was suppressed. Notably, the plasma levels of EETs and DHETs correlated with the Ang II concentration in COVID-19 patients. These findings emphasize the potential value of evaluating biomarkers related to the RAS and ArA pathway for early detection of high-risk groups, prompt medical attention, effective resource allocation, and improved clinical outcomes, ultimately aiming to prevent the occurrence of long COVID.

This study provides a foundation for future research and can serve as a basis for designing more comprehensive studies. Overall, evaluating and including the RAS and ArA pathway biomarkers in decision strategies could provide early detection of high-risk groups, avoid delayed medical attention, facilitate resource allocation, and improve clinical outcomes in COVID-19 patients.

## Figures and Tables

**Figure 1 biomedicines-11-02118-f001:**
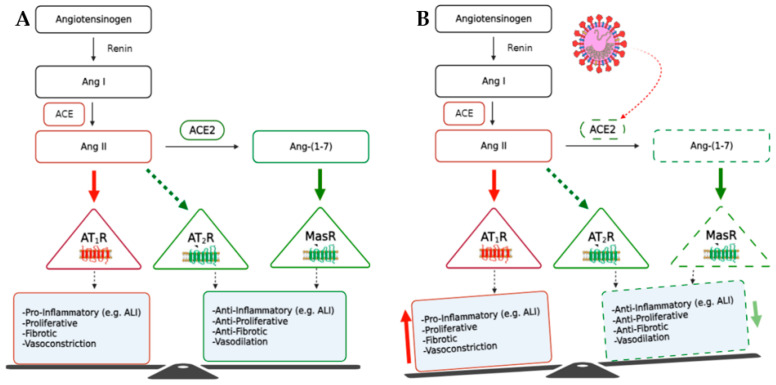
The RAS balance (**A**) and the impact of SARS-CoV-2 on ACE2 protein leading to RAS imbalance (**B**).

**Figure 2 biomedicines-11-02118-f002:**
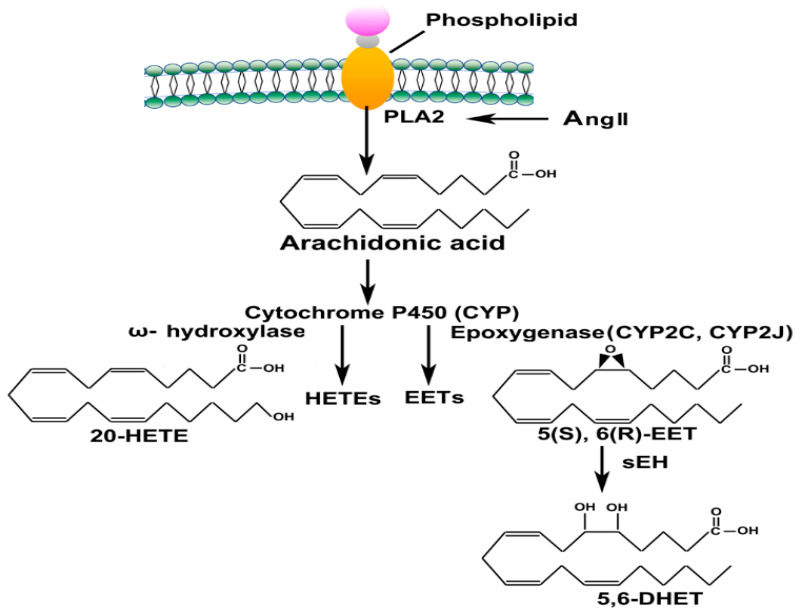
Liberation of ArA and its metabolism by CYP 450 enzymes to hydroxyeicosatetraenoic acids (HETEs), epoxyeicosatrienoic acids (EETs) and dihydroxyeicosatrienoic acids (DHET).

**Figure 3 biomedicines-11-02118-f003:**
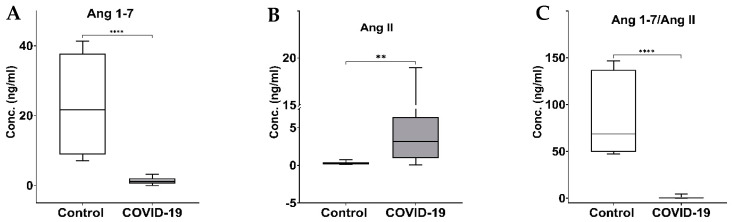
The boxplot of plasma concentration of Ang 1–7 (**A**), Ang II (**B**), and their ratio (**C**) in control (*n* = 6) and COVID-19 (*n* = 24–27). Statistical analysis was performed using the Mann–Whitney U test. Significantly different at ** *p* < 0.001 and **** *p* < 0.0001.

**Figure 4 biomedicines-11-02118-f004:**
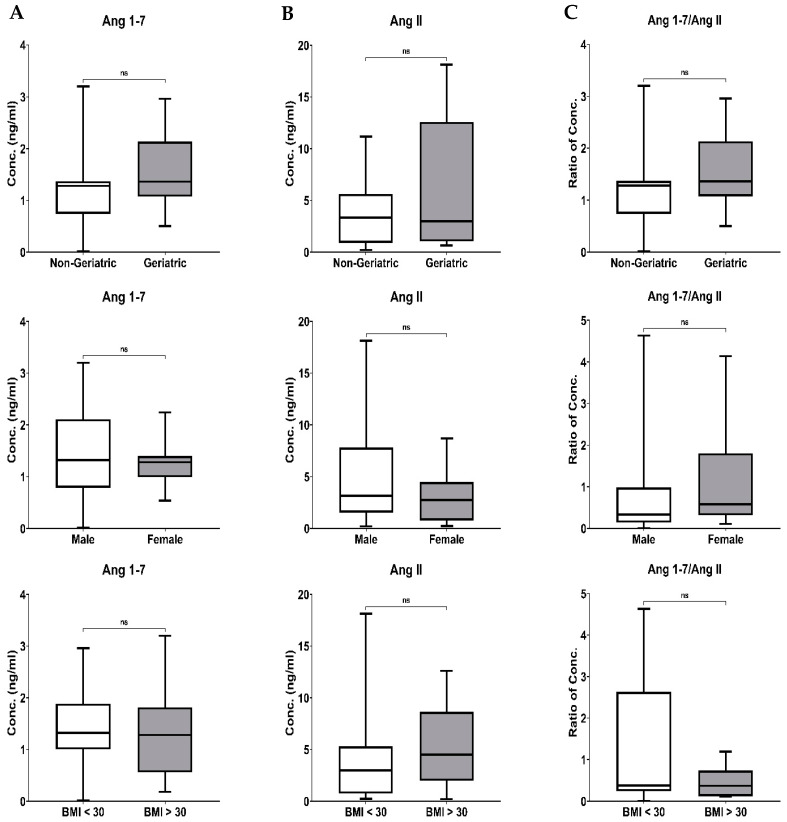
The Boxplots of the impact of age, sex and BMI on the plasma concentration of Ang 1–7 (**A**), Ang II (**B**) and their ratio (Ang 1–7/Ang II) (**C**). Statistical analysis was performed using the Mann–Whitney U test.

**Figure 5 biomedicines-11-02118-f005:**
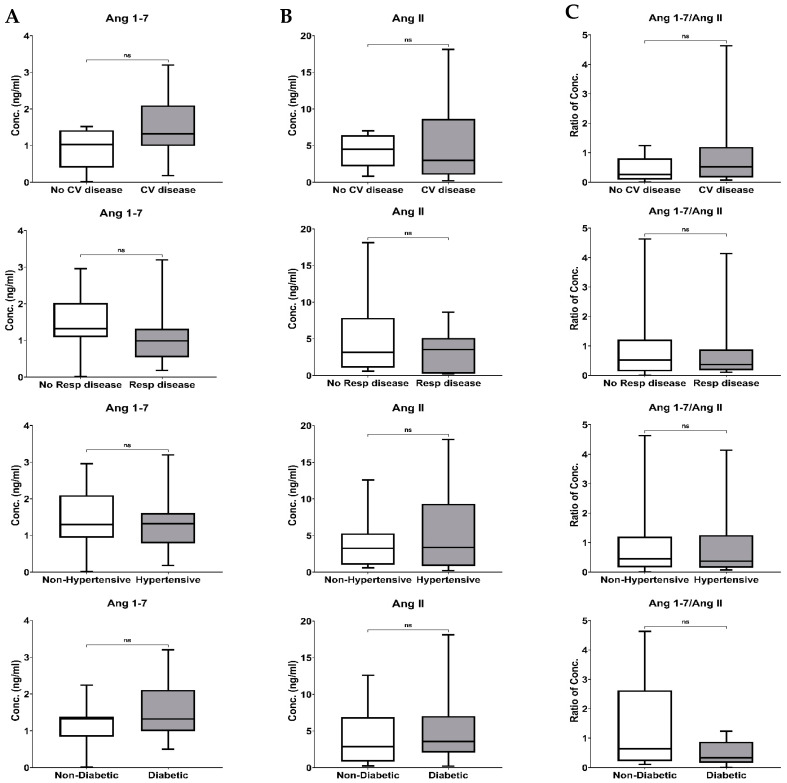
The Boxplots of the impact of comorbidities (respiratory, cardiovascular, hypertension or diabetes) on the plasma concentration of Ang 1–7 (**A**), Ang II (**B**) and their ratio (Ang 1–7/Ang II) (**C**). Statistical analysis was performed using the Mann–Whitney U test.

**Table 1 biomedicines-11-02118-t001:** Clinical characteristics, comorbidities, and medication history of COVID-19 patients.

Clinical Characteristics	
Age (years), Mean ± SD	58.74 ± 12.15
Male (%)	16 (56.26)
BMI, mean ± SD	31.92 ± 8.39
Comorbidities	
Cardiovascular, n (%)	22 (81.48)
– Diabetes (Type I or II)	15 (55.56)
– Ischemic Heart Disease	1 (3.70)
– History of MI	1 (3.70)
– Cardiac Arrhythmia	4 (14.81)
– Hypertension	13 (48.15)
– Valvular Heart Disease	1 (3.70)
– Cerebrovascular disease (Stroke or TIA)	2 (7.41)
Respiratory	
– COPD	2 (7.41)
– Asthma	6 (22.22)
– Other Chronic lung diseases	2 (7.41)
Medication History	
– Aspirin (any dose)	4 (14.81)
– NSAIDs	5 (18.52)
– Statin	10 (37.04)
– Corticosteroids	2 (7.41)
SOFA, Mean ± SD	5.29 ± 2.91
Platelets (per μL), Mean ± SD	241.80 ± 73.41 × 10^3^
Total Bilirubin (mg/dL), Mean ± SD	0.93 ± 0.63
Serum Creatinine (mg/dL), Mean ± SD	1.03 ± 0.61
WBC (per μL), mean ± SD	10.45 ± 5.52 × 10^3^
Hemoglobin (g/dL), Mean ± SD	13.94 ± 1.84

BMI, body mass index; MI, myocardial infarction; TIA, transient ischemic attack; COPD, Chronic obstructive pulmonary disease; NSAID, non-steroidal anti-inflammatory drugs; SOFA, sequential organ failure assessment.

**Table 2 biomedicines-11-02118-t002:** The ArA metabolites levels in Control and COVID-19 patients.

Metabolites	Control	N	COVID-19	*p*	N
19-HETE	2.68 ± 0.89	6	6.35 ± 1.52	0.5605	24
20-HETE	0.99 ± 0.30	6	1.13 ± 0.14	0.5953	24
Total-HETE	3.67 ± 1.11	6	7.49 ± 1.47	0.1286	24
5,6-EET	3.05 ± 0.55	6	13.78 ± 8.06	0.2962	24
8,9-EET	2.87 ± 0.85	6	14.25 ± 2.36	**0.0108**	24
11,12-EET	1.96 ± 0.47	6	41.93 ± 7.79	**<0.0001**	24
14,15-EET	1.95 ± 0.13	6	13.74 ± 8.05	**0.0034**	24
Total-EET	9.83 ± 0.84	6	71.05 ± 12.64	**0.0004**	24
5,6-DHET	7.84 ± 0.90	6	1.98 ± 0.39	**<0.0001**	16
8,9-DHET	3.85 ± 0.36	6	3.04 ± 0.43	**0.0252**	19
11,12-DHET	3.58 ± 0.63	6	0.74 ± 0.21	**0.0004**	9
14,15-DHET	4.28 ± 0.48	6	1.42 ± 0.18	**0.0001**	24

Data are presented as Mean ± SEM and expressed in ng/mL. Statistical analysis was performed using the Mann–Whitney U test.

**Table 3 biomedicines-11-02118-t003:** The correlation of Ang 1–7 with ArA metabolites in Control and COVID-19 patients.

Metabolites	Ang 1–7					
	Control			COVID-19		
	r	*p*	N	r	*p*	N
19-HETE	0.4294	0.3955	6	−0.1773	0.4071	24
20-HETE	0.5718	0.2358	6	**−0.4089**	**0.0473**	24
Total-HETE	0.3417	0.5075	6	−0.2231	0.2947	24
5,6-EET	0.2850	0.5841	6	−0.0391	0.8560	24
8,9-EET	0.7324	0.0978	6	−0.1328	0.5362	24
11,12-EET	−0.1309	0.8048	6	−0.1765	0.4094	24
14,15-EET	0.0110	0.9836	6	−0.0507	0.8141	24
Total-EET	**0.8559**	**0.0296**	6	−0.1854	0.3858	24
5,6-DHET	0.2511	0.6312	6	−0.3823	0.1439	16
8,9-DHET	−0.4195	0.4077	6	−0.0314	0.8985	19
11,12-DHET	0.1751	0.7401	6	−0.4462	0.2286	9
14,15-DHET	−0.1969	0.7085	6	−0.2836	0.1793	24

Statistical analysis was performed using the Pearson correlation coefficient (r) computation.

**Table 4 biomedicines-11-02118-t004:** The correlation of Ang II with ArA metabolites in Control and COVID-19 patients.

Metabolites	Ang II					
	Control			COVID-19		
	r	*p*	N	r	*p*	N
19-HETE	0.6614	0.1525	6	**0.5052**	**0.0195**	24
20-HETE	0.5550	0.2530	6	0.2871	0.2333	24
Total-HETE	0.6797	0.1375	6	**0.5317**	**0.0131**	24
5,6-EET	0.7516	0.0849	6	**0.6805**	**0.0007**	24
8,9-EET	−0.0180	0.9729	6	0.1021	0.6595	24
11,12-EET	−0.5656	0.2421	6	0.0377	0.8712	24
14,15-EET	0.5736	0.2339	6	**0.6721**	**0.0008**	24
Total-EET	0.2392	0.6481	6	**0.5998**	**0.0041**	24
5,6-DHET	**0.8401**	**0.0363**	6	−0.0191	0.9843	14
8,9-DHET	−0.0445	0.9333	6	**0.6290**	**0.0068**	17
11,12-DHET	0.5164	0.2943	6	0.2578	0.5376	8
14,15-DHET	0.0947	0.8584	6	0.0244	0.9165	24

Statistical analysis was performed using the Pearson correlation coefficient (r) computation.

**Table 5 biomedicines-11-02118-t005:** The correlation of Ang 1–7/Ang II with ArA metabolites in Control and COVID-19 patients.

Metabolites	Ang 1–7/Ang II					
	Control			COVID-19		
	r	*p*	N	r	*p*	N
19-HETE	0.0626	0.9062	6	−0.2539	0.2668	21
20-HETE	0.1606	0.7612	6	−0.3434	0.1275	21
Total-HETE	0.0934	0.8603	6	−0.2959	0.1928	21
5,6-EET	−0.4236	0.4027	6	−0.1546	0.5034	21
8,9-EET	**0.8589**	**0.0285**	6	−0.1683	0.4659	21
11,12-EET	0.2500	0.6328	6	−0.0802	0.7298	21
14,15-EET	−0.4992	0.3134	6	−0.1552	0.5019	21
Total-EET	0.6601	0.1537	6	−0.1943	0.3987	21
5,6-DHET	−0.5780	0.2296	6	−0.2551	0.3788	14
8,9-DHET	−0.3861	0.4497	6	−0.1421	0.5863	17
11,12-DHET	−0.0865	0.8706	6	−0.4886	0.2192	8
14,15-DHET	−0.0757	0.8867	6	−0.2089	0.3635	21

Statistical analysis was performed using the Pearson correlation coefficient (r) computation.

**Table 6 biomedicines-11-02118-t006:** The correlation of the RAS components with ArA metabolites levels in all participants.

Metabolites	Ang 1–7		Ang II		Ang 1–7/Ang II	
	r	*p*	r	*p*	r	*p*
19-HETE	−0.1764	0.3989	−0.1652	0.3919	−0.2130	0.3067
20-HETE	0.0558	0.7912	0.1931	0.3156	−0.0391	0.8528
Total-HETE	−0.1689	0.4195	−0.1380	0.4753	−0.2181	0.2950
5,6-EET	−0.1398	0.5051	−0.0812	0.6753	−0.1652	0.4301
8,9-EET	−0.2475	0.2794	0.0406	0.8472	−0.2469	0.2807
11,12-EET	**−0.3899**	**0.0597**	−0.0526	0.7902	−0.3855	0.0628
14,15-EET	−0.2764	0.1810	−0.1143	0.5550	−0.2651	0.2003
Total-EET	**−0.3874**	**0.0557**	−0.0874	0.6523	−0.3805	0.0606
5,6-DHET	**0.7496**	**0.0002**	−0.3568	0.1124	**0.6617**	**0.0020**
8,9-DHET	0.4141	0.0695	0.2052	0.3476	**0.4651**	**0.0388**
11,12-DHET	**0.6704**	**0.0122**	−0.2438	0.4010	**0.6368**	**0.0192**
14,15-DHET	**0.5930**	**0.0018**	−0.1370	0.4786	**0.6684**	**0.0003**

Statistical analysis was performed using the Pearson correlation coefficient (r) computation.

**Table 7 biomedicines-11-02118-t007:** The correlation of the SOFA score with the RAS components and ArA metabolites levels in COVID-19 patients.

SOFA Score		
Metabolites	r	*p*
Ang 1–7	0.1825	0.3723
Ang II	**0.4171**	**0.0426**
Ang 1–7/Ang II	0.2814	0.1933
19-HETE	**−0.4842**	**0.0165**
20-HETE	0.1967	0.3568
Total-HETE	**−0.4828**	**0.0169**
5,6-EET	−0.2872	0.1735
8,9-EET	−0.0410	0.8492
11,12-EET	−0.2100	0.3247
14,15-EET	−0.2758	0.1920
Total-EET	−0.3882	0.0646
5,6-DHET	−0.3095	0.2435
8,9-DHET	−0.3773	0.1112
11,12-DHET	−0.4211	0.2590
14,15-DHET	−0.0515	0.8111

Statistical analysis was performed using the Pearson correlation coefficient (r) computation.

## Data Availability

The data presented in this study are available on request from the corresponding author.

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
