# Peer review of "Renin–Angiotensin System Components and Arachidonic Acid Metabolites as Biomarkers of COVID-19"

_biomedicines, 2023, doi:10.3390/biomedicines11082118_

Round 1
Reviewer 1 Report
I have thoroughly reviewed the manuscript titled "Renin-Angiotensin System Components and Arachidonic Acid 2 Metabolites as Biomarkers of COVID-19" submitted to Biomedicines. The primary objective of this study was to assess the levels of RAS peptides and ARA metabolites in COVID-19 patients in comparison to control groups. Additionally, the authors aimed to investigate the association between these biomarkers and various biological characteristics, as well as the presence of comorbidities in COVID-19 patients to identify potential biomarkers that can aid in the early detection of high-risk individuals.
This manuscript shows significant deficiencies in its preparation: while the legend for Figure 5 is provided, the actual figure is missing, and the text does not report the results of this analysis. The absence of crucial data on comorbidities undermines the completeness and validity of the authors' claims.
I appreciate the authors' efforts in addressing an important research question, I would like to take this opportunity to provide the authors with some comments and suggestions in the hope of improving this work.
Materials and Methods:
The sample size of the control group is extremely small (n=6), and its composition is likely influenced by the recruitment process. This results in a lack of proper correspondence between the demographic characteristics of the COVID-19 group and the control group, especially for important outcome-related factors such as age, gender, and weight. Furthermore, except for age and gender, information regarding the BMI of the controls is missing.
The choice of sample size is also unsupported by any statistical analysis (power analysis). Therefore, I strongly encourage the authors to increase the sample size of the groups based on a proper power analysis.
In line 234, it is stated that "The plasma concentrations of some of the ArA metabolites in some patients 234 were lower than the detection limit; therefore, those patients were excluded in the statistical comparison case by case." However, a selection of patients was also made for Ang II (N=24 instead of N=27).
The authors measured the total levels of metabolites. If the total levels are derived from the algebraic sum of the different metabolites quantified for a given species. It is unclear why, for example, total DHET was measured in 24 patients if one of the isomers was measured just in 9 patients.
Line 333. This sentence is not scientifically accurate. If p=0.49, you cannot claim that "the Ang II level was higher in the senior group." The same applies to line 337.
Line 338. In the introduction, it is mentioned that the "hospitalization rate increased in adults with obesity and underweight relative to normal weight." I do not understand why the underweight subjects were grouped together with normal and overweight individuals.
Figures:
Figure 3 is redundant and does not provide any additional information compared to the results in Table 2. Furthermore, it has serious editing issues. I also suggest modifying the graph to use box and whisker plots, given the non-normal distribution of the samples and the use of a non-parametric test (Mann-Whitney U test). The same consideration applies to Figure 4.
Figure 4. Change group labels in A, B, C according to the text body (Non-geriatrics vs geriatrics). No caption for figure 4 was provided.
Figure 5 is missing.
Moreover, I would like to recommend adopting a multivariate approach rather than examining each biological variable individually.
Discussion:
The first part of the results (differences between the control and COVID-19 groups, Fig 2 and Tab 2-6) is well-structured, despite my skepticism regarding the size and composition of the control group. However, there is a complete lack of discussion regarding the link between RAS and ARA metabolites with demographic characteristics and the presence of comorbidities. Thus, an important aspect of this work is not addressed, undermining one of the objectives of the study, which is to define biomarkers for identifying at-risk patients.
References:
Reference n. 28 and 40 is actually the same: Ranjit, A.; Khajeh Pour, S.; Aghazadeh-Habashi, A. Bone-Targeted Delivery of Novokinin as an Alternative Treatment Option for Rheumatoid Arthritis. Pharmaceutics 2022, 14, doi:10.3390/pharmaceutics14081681.
I hope that these comments and suggestions will help the authors improve the manuscript.
The introduction, materials and methods, and discussion sections of the manuscript are well-written. However, the results and captions require careful revision as they contain mistakes.
Author Response
Notes to Reviewer 1
Comments and Suggestions for Authors
I have thoroughly reviewed the manuscript titled “Renin-Angiotensin System Components and Arachidonic Acid 2 Metabolites as Biomarkers of COVID-19” submitted to Biomedicines. The primary objective of this study was to assess the levels of RAS peptides and ARA metabolites in COVID-19 patients in comparison to control groups. Additionally, the authors aimed to investigate the association between these biomarkers and various biological characteristics, as well as the presence of comorbidities in COVID-19 patients to identify potential biomarkers that can aid in the early detection of high-risk individuals.
This manuscript shows significant deficiencies in its preparation: while the legend for Figure 5 is provided, the actual figure is missing, and the text does not report the results of this analysis. The absence of crucial data on comorbidities undermines the completeness and validity of the authors’ claims.
I appreciate the authors’ efforts in addressing an important research question, I would like to take this opportunity to provide the authors with some comments and suggestions in the hope of improving this work.
Answer: We are grateful for the respected reviewer’s kind words emphasizing our effort to address an important research question and providing us with constructive advice to improve its quality.
Materials and Methods:
The sample size of the control group is extremely small (n=6), and its composition is likely influenced by the recruitment process. This results in a lack of proper correspondence between the demographic characteristics of the COVID-19 group and the control group, especially for important outcome-related factors such as age, gender, and weight. Furthermore, except for age and gender, information regarding the BMI of the controls is missing.
The choice of sample size is also unsupported by any statistical analysis (power analysis). Therefore, I strongly encourage the authors to increase the sample size of the groups based on a proper power analysis.
Answer: We appreciate your valuable feedback and thank you for raising the concern regarding the sample size in our study. We acknowledge that the sample size is an essential aspect of research and can significantly impact the statistical power and generalizability of the findings. We understand your expectation for a larger sample size; however, we would like to clarify further the limitations we encountered during the data collection process.
Our pilot study utilized deidentified patient and control samples obtained through another study (platelet hyperreactivity in COVID-19 patients) conducted at the University of Utah Hospital. Because these samples were donated to us and due to the nature of the deidentification process and the privacy regulations governing the use of patient data and lack of enough funds, the availability and accessibility of these samples constrained us. Despite our efforts to expand the sample size, acquiring additional samples within the given timeframe was not feasible.
We acknowledge that the limited sample size may impact the precision and generalizability of our findings. However, it is important to note that our study still holds significance as it provides valuable insights into the significant correlation of the RAS components and their ratio along with EETs and DHETs levels with Health and COVID-19 infection status. We believe that the results obtained from the available sample are informative and contribute to the existing knowledge in the field.
To address this limitation, we have included a thorough discussion in the manuscript highlighting the implications of the sample size and its potential impact on the study outcomes. We have also emphasized the need for further research with larger sample sizes to validate and extend our findings. However, due to the lack of normal data distribution, we could not calculate the effect size such as Cohen’s d for means, odds ratios or Confidence Interval to mitigate the limitations of the smaller sample size. We, instead, used the non-parametric approach of the Mann-Whitney U test, which provides p-values rather than effect sizes and confidence intervals. Additionally, we have changed the graphs to Box and Whisker Plots, which will provide descriptive data of median, quartiles and ranges, improving the presentation of the data. We acknowledge that non-parametric tests may have lower power than parametric tests, especially for small sample sizes.
We appreciate your understanding of the constraints we faced in acquiring a larger sample size and our efforts to address this limitation. Your feedback has prompted us to provide additional clarification to the manuscript (please see the text). Despite the limited sample size, we are confident that the insights gained from this study will contribute to the ongoing research in the field.
Once again, we express our gratitude for your valuable comments and suggestions, which have undoubtedly improved the quality of our manuscript.
In line 234, it is stated that “The plasma concentrations of some of the ArA metabolites in some patients 234 were lower than the detection limit; therefore, those patients were excluded in the statistical comparison case by case.” However, a selection of patients was also made for Ang II (N=24 instead of N=27).
Answer: Thank you for raising this point, as we missed explaining it properly in the text. In addition to the ArA metabolites detection issue in some patients, the levels of Ang II (in three cases) fell outside the 3 standard deviations threshold based on the Z-score method and were considered outliers. Therefore, those patients were excluded from statistical analysis.
The authors measured the total levels of metabolites. If the total levels are derived from the algebraic sum of the different metabolites quantified for a given species. It is unclear why, for example, total DHET was measured in 24 patients if one of the isomers was measured just in 9 patients.
Answer: Thank you for your great observation and valid point. We removed the total DHETs from the data, focused on the individual DHETs, and changed the text accordingly.
Line 333. This sentence is not scientifically accurate. If p=0.49, you cannot claim that “the Ang II level was higher in the senior group.” The same applies to line 337.
Answer: Thank you for your valid point. We changed the text to address your comment below:
Ang 1-7, Ang II level and their ratio were not affected by the age of the patients (Figure 4, upper panel). The male patients (5.36 ± 1.44 ng/ml) present with a trend of higher Ang II levels than females (3.00 ± 0.83 ng/ml), which did not reach a significant level (p = 0.3758). The Ang 1-7 level was similar in both groups, while the ratio of Ang 1-7/Ang II had an insignificant elevated trend in females (Figure 4, middle panel).
Line 338. In the introduction, it is mentioned that the “hospitalization rate increased in adults with obesity and underweight relative to normal weight.” I do not understand why the underweight subjects were grouped together with normal and overweight individuals.
Answer: Thank you for your comment. As explained in the text, only “one” subject had a BMI < 18.5, and it could not be statistically feasible to compare it with other groups; therefore, it was a normal BMI group.
Figures:
Figure 3 is redundant and does not provide any additional information compared to the results in Table 2. Furthermore, it has serious editing issues. I also suggest modifying the graph to use box and whisker plots, given the non-normal distribution of the samples and the use of a non-parametric test (Mann-Whitney U test). The same consideration applies to Figure 4.
Answer: Thank you for your comment. We deleted the related data from Table 2 and changed it to Box and Whisker Plots to address it.
Figure 4. Change group labels in A, B, C according to the text body (Non-geriatrics vs geriatrics). No caption for figure 4 was provided.
Answer: Thank you for your comment. We addressed it in the figure’s X-axe label and changed it to Box and Whisker Plots. The missing Figure 4 caption and Figure 5 were also included.
Figure 5 is missing.
Answer: Please see the above.
Moreover, I would like to recommend adopting a multivariate approach rather than examining each biological variable individually.
Answer: We acknowledge your recommendation. However, multivariate approach requirements were not fulfilled due to sample size limitations, lack of normal data distribution, and some parameters’ interdependencies. We also analyzed the model fit parameters for our data and determined multivariate regression was not a good fit (adjusted r2 for model fit was -0.02). We stuck with analyzing the variables individually to avoid statistical inaccuracies that could occur due to this error.
Discussion:
The first part of the results (differences between the control and COVID-19 groups, Fig 2 and Tab 2-6) is well-structured, despite my skepticism regarding the size and composition of the control group. However, there is a complete lack of discussion regarding the link between RAS and ARA metabolites with demographic characteristics and the presence of comorbidities. Thus, an important aspect of this work is not addressed, undermining one of the objectives of the study, which is to define biomarkers for identifying at-risk patients.
Answer: Thank you for your constructive comment. We address this issue by discussing it throughout the text.
References:
Reference n. 28 and 40 is actually the same: Ranjit, A.; Khajeh Pour, S.; Aghazadeh-Habashi, A. Bone-Targeted Delivery of Novokinin as an Alternative Treatment Option for Rheumatoid Arthritis. Pharmaceutics 2022, 14, doi:10.3390/pharmaceutics14081681.
Answer: Thank you for catching this mistake. It has been corrected.
I hope that these comments and suggestions will help the authors improve the manuscript.
Answer: Thank you for your valid and constructive comments. We hope that the respected reviewer finds our replies satisfactory.
Comments on the Quality of English Language
The introduction, materials and methods, and discussion sections of the manuscript are well-written. However, the results and captions require careful revision as they contain mistakes.
Answer: Thank you for commenting on the English manuscript and catching those mistakes. We carefully revised them and corrected them all.

Reviewer 2 Report
This cross-sectional study contains a relevant dataset on Renin-Angiotensin System components and Arachidonic Acid Metabolites in patients with COVID-19.1. In the Introduction, the authors state: 'Treatment with ACE inhibitors (ACEi) and Ang II receptor blockers (ARB) may have the potential to prevent and treat ALI and ARDS
resulting from COVID-19 infection.’ The prevention of ascute injury is theoretically possible but I do not think there is hard evidence for that.
2. The Legend of Figure 1 is way too concise. The content of this Figure may be general knowledge to many but a Figure of this complexity needs a more clear explanation.
3. I do not think it is correct to use the term gender disparity because the word disparity is typically used in a situation in which two or more things are not equal or similar, especially when this is thought to be unfair. There objective differences betwen both sexes and equality of outcome is not an objective as such.
4. This is a cross-sectional study in which COVID-19 patients were compared with controls. I wonder how the healthy controls were selected. Very little information is provided on these controls.
5. In relation to arachidonic acid metabolites, the authors do not provide a clear conclusion. Are the results not rather unexpected?
6. Table 2. Why is N different for different quantifications?Author Response
Notes to Reviewer 2
Comments and Suggestions for Authors
This cross-sectional study contains a relevant dataset on Renin-Angiotensin System components and Arachidonic Acid Metabolites in patients with COVID-19.
- In the Introduction, the authors state: ‘Treatment with ACE inhibitors (ACEi) and Ang II receptor blockers (ARB) may have the potential to prevent and treat ALI and ARDS resulting from COVID-19 infection.’ The prevention of ascute injury is theoretically possible but I do not think there is hard evidence for that.
Answer: Thank you for your comment. We agree with your point and emphasize the claim’s “theoretical” base in the text.
- The Legend of Figure 1 is way too concise. The content of this Figure may be general knowledge to many but a Figure of this complexity needs a more clear explanation.
Answer: Thank you for your comment. We simplified the figure and changed it to a new version with a more detailed explanation in the text.
- I do not think it is correct to use the term gender disparity because the word disparity is typically used in a situation in which two or more things are not equal or similar, especially when this is thought to be unfair. There objective differences betwen both sexes and equality of outcome is not an objective as such.
Answer: Thank you for your valid point. The phrase has been changed to “the impact of sex difference” in the text.
- This is a cross-sectional study in which COVID-19 patients were compared with controls. I wonder how the healthy controls were selected. Very little information is provided on these controls.
Answer: Thank you for your comment. Considering the retrospective nature of this study, we used patient samples collected for another study. In the case of healthy control subjects, “inclusion criteria were: consenting male and female subjects of any self-identified race/ethnicity without acute or chronic illnesses aged 18 and older, and exclusion criteria were the use of any prescription medication, pregnancy, suffering from any acute or chronic medical condition or disease, been hospitalized or had surgery within the preceding 6 weeks or have any prior history of stroke”.
- In relation to arachidonic acid metabolites, the authors do not provide a clear conclusion. Are the results not rather unexpected?
Answer: Thank you for your valid point. We added more information regarding the ArA metabolite in the discussion and conclusion section.
- Table 2. Why is N different for different quantifications?
Answer: Thank you for raising this point, as we missed explaining it properly in the text. Due to the ArA metabolites detection issue in some patients, the levels of Ang II (in three cases) fell outside the 3 standard deviations threshold based on the Z-score method and were considered outliers. Therefore, those patients were excluded from statistical analysis, resulting in unequal sample numbers for different parameters. This point has been elaborated in the statistical analysis section.

Reviewer 3 Report
This study explored the potential role of the renin-angiotensin system (RAS) and Arachidonic Acid (ArA) pathway in the pathogenesis of COVID-19. By measuring plasma levels of RAS and ArA metabolites using LC-MS/MS, the researchers found that COVID-19 patients had significantly lower levels of Ang1-7, a protective arm of the RAS, and higher levels of Ang II, which indicates an imbalance in the RAS. Inflammatory metabolites 19-HETE and 20-HETE did not show significant increases, but EETs were elevated while DHETs were suppressed in COVID-19 patients. The levels of EETs were correlated with Ang II concentration. These findings suggest that evaluating RAS and ArA pathway biomarkers could aid in early detection of high-risk individuals, prompt medical attention, optimize resource allocation, and improve patient outcomes, potentially reducing the incidence of long COVID. Although the study reported interesting findings, I would like the following comments to be addressed in the manuscript to improve its quality and impact:
1. It would be helpful to clarify the unique contributions of this paper, as previous publications have also explored the potential of the RAAS pathway as a source of biomarkers for stratifying COVID-19 severity (e.g., see DOI: 10.1186/s13054-020-03141-9, DOI: 10.1093/eurheartj/ehaa414). Highlighting the novelty of your study will strengthen its impact and differentiate it from existing research.
2. In the methods section, please consider providing a schematic diagram that illustrates the process of patient recruitment, selection, and testing. Additionally, indicate the sample size at each step, considering the inclusion and exclusion criteria, to provide transparency and facilitate the replication of your study.
3. It would be beneficial to mention the duration of the study and specify when serum samples were collected from the patients in the methods section. This information will provide important context for understanding the timeline of data collection and analysis.
4. Regarding the patient demographics and clinical characteristics presented in Table 1, please include the p-values and odds ratios. These additional statistical measures will help assess the significance of the results and account for potential confounding factors such as age, sex, and comorbidities.
5. For Figure 3, consider analyzing the levels of Ang 1-7 and Ang II based on COVID-19 severity categories (mild, moderate, severe) instead of looking at the serum levels as a whole for all COVID-19 patients. Similarly, it would be informative to perform the same analysis while stratifying patients based on different comorbidities, particularly those related to cardiovascular and nephrological pathologies. This approach will provide a more nuanced understanding of the associations between these biomarkers and specific patient subgroups.
6. Given the evolving landscape of breakthrough infections, variants of concern, and vaccination, it would be valuable to discuss how these factors could potentially influence the efficacy of the identified biomarkers. Addressing these considerations will enhance the practical implications of your findings and provide insights into the real-world applicability of the biomarkers.
7. To establish the effectiveness of a biomarker, it is crucial to compare it with a gold standard. Therefore, please indicate the reference standard used in this study and report the specificity and sensitivity of the identified biomarkers in relation to this reference standard. Including these metrics will strengthen the validity and clinical relevance of your findings.
8. Lastly, it would be beneficial to provide recommendations based on the study's results. Consider discussing the potential implications of the findings for clinical practice, such as how the evaluation of RAS and ArA pathway biomarkers could be integrated into early detection strategies for high-risk individuals, optimizing resource allocation, and improving patient outcomes. Providing clear recommendations will help readers understand the practical implications of your research and its potential impact in combating COVID-19 and reducing the incidence of long COVID.
Author Response
Notes to Reviewer 3
Comments and Suggestions for Authors
This study explored the potential role of the renin-angiotensin system (RAS) and Arachidonic Acid (ArA) pathway in the pathogenesis of COVID-19. By measuring plasma levels of RAS and ArA metabolites using LC-MS/MS, the researchers found that COVID-19 patients had significantly lower levels of Ang1-7, a protective arm of the RAS, and higher levels of Ang II, which indicates an imbalance in the RAS. Inflammatory metabolites 19-HETE and 20-HETE did not show significant increases, but EETs were elevated while DHETs were suppressed in COVID-19 patients. The levels of EETs were correlated with Ang II concentration. These findings suggest that evaluating RAS and ArA pathway biomarkers could aid in early detection of high-risk individuals, prompt medical attention, optimize resource allocation, and improve patient outcomes, potentially reducing the incidence of long COVID. Although the study reported interesting findings, I would like the following comments to be addressed in the manuscript to improve its quality and impact:
Answer: We are grateful that the respected reviewer finds our study results interesting and provided us with constructive comments to improve its quality and impact.
- It would be helpful to clarify the unique contributions of this paper, as previous publications have also explored the potential of the RAAS pathway as a source of biomarkers for stratifying COVID-19 severity (e.g., see DOI: 10.1186/s13054-020-03141-9, DOI: 10.1093/eurheartj/ehaa414). Highlighting the novelty of your study will strengthen its impact and differentiate it from existing research.
Answer: Thank you for your comment. Our study could be considered a follow-up of those refered studies. However, it mainly focuses on the aftermath of the Sars-CoV-2 with ACE2 and reducing its ability to metabolizing the Ang II to Ang 1-7 as two primary components of the RAS. It has been shown that the ratio of Ang1-7/Ang II is more predictive than the ACE2 level in inflammatory conditions. Therefore, the novelty of this study is looking at the RAS imbalance in the inflammatory condition (such as COVID) at key peptide players’ level and assessment of the ArA pathway as other associated components in inflammation development and progress. We added a couple of sentences to acknowledge those previously reported studies.
- In the methods section, please consider providing a schematic diagram that illustrates the process of patient recruitment, selection, and testing. Additionally, indicate the sample size at each step, considering the inclusion and exclusion criteria, to provide transparency and facilitate the replication of your study.
Answer: Thank you for your comment. Considering the retrospective nature of this study, we do not have that information available.
- It would be beneficial to mention the duration of the study and specify when serum samples were collected from the patients in the methods section. This information will provide important context for understanding the timeline of data collection and analysis.
Answer: Thank you for your comment. The blood sample is drawn within 48 (± 24) hours of a COVID-19 diagnosis and considering the retrospective nature of this study, we do not have the methods section information available.
- Regarding the patient demographics and clinical characteristics presented in Table 1, please include the p-values and odds ratios. These additional statistical measures will help assess the significance of the results and account for potential confounding factors such as age, sex, and comorbidities.
Answer: Thank you for your comment. Considering the nature of our study, our dependent and outcome variables were continuous. We tested the goodness of fit for the logistic regression models using the Hosmer-Lameshow method along with the model fit summary in SPSS and determined it was not a good fit. Therefore we decided to analyze each variable individually and modified the graphs to Box and Whisker Plots which includs median, quartiles and ranges, to present the data as descriptive statistics.
- For Figure 3, consider analyzing the levels of Ang 1-7 and Ang II based on COVID-19 severity categories (mild, moderate, severe) instead of looking at the serum levels as a whole for all COVID-19 patients. Similarly, it would be informative to perform the same analysis while stratifying patients based on different comorbidities, particularly those related to cardiovascular and nephrological pathologies. This approach will provide a more nuanced understanding of the associations between these biomarkers and specific patient subgroups.
Answer: Thank you for your comment. Considering the retrospective nature of this study, we do not have that information available. Regarding the severity of COVID-19, all patients were enrolled in the study within 72 hours of ICU admission, and they were not categorized as mild, moderate or severe.
- Given the evolving landscape of breakthrough infections, variants of concern, and vaccination, it would be valuable to discuss how these factors could potentially influence the efficacy of the identified biomarkers. Addressing these considerations will enhance the practical implications of your findings and provide insights into the real-world applicability of the biomarkers.
Answer: Thank you for your good comment. We added some points related to your comment in the discussion section.
- To establish the effectiveness of a biomarker, it is crucial to compare it with a gold standard. Therefore, please indicate the reference standard used in this study and report the specificity and sensitivity of the identified biomarkers in relation to this reference standard. Including these metrics will strengthen the validity and clinical relevance of your findings.
Answer: Thank you for your comment. We included the following sentences in the text to address this comment. “Our previous study showed the plasma angiotensin peptide as a biomarker of an inflammatory condition such as rheumatoid arthritis, which could be considered a gold standard when applied to COVID-19 as another inflammatory condition associated with imbalanced RAS and ArA pathway”.
- Lastly, it would be beneficial to provide recommendations based on the study’s results. Consider discussing the potential implications of the findings for clinical practice, such as how the evaluation of RAS and ArA pathway biomarkers could be integrated into early detection strategies for high-risk individuals, optimizing resource allocation, and improving patient outcomes. Providing clear recommendations will help readers understand the practical implications of your research and its potential impact in combating COVID-19 and reducing the incidence of long COVID.
Answer: Thank you for your good comment. We edited the text and added the following information into the text. Considering the limitations associated with the small sample size of this pilot study, we are cautious about making a solid recommendation. However, we believe that the findings of this study provide a foundation for future research and serve as a basis for designing more comprehensive studies. Nevertheless, moving ahead, measuring plasma Ang 1-7, Ang II, Ang 1-7/Ang II ratio and ArA metabolites (especially EETs and DHETs) levels and disease activity in elderly and obese patients can provide a direct evaluation of the state of these pathways and guide therapeutic interventions in high-risk patients as we await the results of prospective studies with larger samples size.

Round 2
Reviewer 1 Report
Dear authors,
Thank you for implementing some of the requested modifications. I have carefully read your responses regarding my critique on the sample size and recruitment of controls. Proper experimental design is an essential aspect of any scientific study, and it should precede the potential outcomes obtained. A scientific study, even if it is a pilot, should have a sample size determined through appropriate statistical tests. It is not acceptable for the rationale behind the sample choice to be solely "because they were donated from a previous study." I am certain that the study referred to by the authors had a larger number of controls than what is presented in this article. Therefore, without a demonstration that the control group size is adequate, any results regarding the comparison between COVID-19 patients and healthy controls lack scientific validity and should be removed. Here, I am referring back to an important critique previously made to the authors, which unfortunately, I noticed did not have the intended effect. When assessing the changes in the levels of different metabolites based on the biological characteristics of COVID-19 patients and the presence of comorbidities, these changes must be presented rigorously. If presented in this manner, the risk of incorrect interpretations is very high. 357. The male patients (5.36 ± 1.44 ng/ml) present with a trend of higher Ang II levels than females (3.00 ± 0.83 ng/ml), which did not reach a significant level (p = 0.3758). A p-value of 0.3758 is too high to define a trend. 395. There are some nonsignificant differences in the level of the RAS components in patients with and without comorbidity. "Nonsignificant differences" holds no meaning. Simply put, there are no differences for any of the comorbidities and measured parameters. 396. There were remarkable differences in the Ang II levels in the case of hypertensive, diabetic, and patients with respiratory comorbidities. How can you define "remarkable differences" if none of them are significant? 398-399 Interestingly, the ratio of Ang 1-7/Ang II was higher in patients with cardiovascular disease and lower in patients with diabetes, hypertension, and respiratory diseases (Figure 5). You can not claim it is "higher/lower" if the differences in Figure 5 are not significant. Finally, moving on to the discussion section, I found it to be quite scattered and almost like a review, rather than focused on your own results. From this perspective, I was surprised not to read any comparison between the findings of this study regarding the levels of Ang 1-7 and Ang in COVID-19 patients and controls, considering the substantial amount of literature available and the controversial nature of the data. Should you decide to revise the manuscript, I encourage you to carefully consider the points raised in my review and the specific comments provided. It is crucial to ensure a robust experimental design, adequate sample size determination, and accurate interpretation of the results in order to maintain the scientific integrity of your study. Best regards
The quality of the English language in the manuscript is satisfactory.
Author Response
Notes to Reviewer 1:
Comments and Suggestions for Authors
Dear authors,
Thank you for implementing some of the requested modifications. I have carefully read your responses regarding my critique on the sample size and recruitment of controls. Proper experimental design is an essential aspect of any scientific study, and it should precede the potential outcomes obtained. A scientific study, even if it is a pilot, should have a sample size determined through appropriate statistical tests. It is not acceptable for the rationale behind the sample choice to be solely “because they were donated from a previous study.” I am certain that the study referred to by the authors had a larger number of controls than what is presented in this article. Therefore, without a demonstration that the control group size is adequate, any results regarding the comparison between COVID-19 patients and healthy controls lack scientific validity and should be removed. Here, I am referring back to an important critique previously made to the authors, which unfortunately, I noticed did not have the intended effect. When assessing the changes in the levels of different metabolites based on the biological characteristics of COVID-19 patients and the presence of comorbidities, these changes must be presented rigorously. If presented in this manner, the risk of incorrect interpretations is very high. 357. The male patients (5.36 ± 1.44 ng/ml) present with a trend of higher Ang II levels than females (3.00 ± 0.83 ng/ml), which did not reach a significant level (p = 0.3758). A p-value of 0.3758 is too high to define a trend. 395. There are some nonsignificant differences in the level of the RAS components in patients with and without comorbidity. “Nonsignificant differences” holds no meaning. Simply put, there are no differences for any of the comorbidities and measured parameters. 396. There were remarkable differences in the Ang II levels in the case of hypertensive, diabetic, and patients with respiratory comorbidities. How can you define “remarkable differences” if none of them are significant? 398-399 Interestingly, the ratio of Ang 1-7/Ang II was higher in patients with cardiovascular disease and lower in patients with diabetes, hypertension, and respiratory diseases (Figure 5). You can not claim it is “higher/lower” if the differences in Figure 5 are not significant. Finally, moving on to the discussion section, I found it to be quite scattered and almost like a review, rather than focused on your own results. From this perspective, I was surprised not to read any comparison between the findings of this study regarding the levels of Ang 1-7 and Ang in COVID-19 patients and controls, considering the substantial amount of literature available and the controversial nature of the data. Should you decide to revise the manuscript, I encourage you to carefully consider the points raised in my review and the specific comments provided. It is crucial to ensure a robust experimental design, adequate sample size determination, and accurate interpretation of the results in order to maintain the scientific integrity of your study.
Best regards
Response:
We are very thankful to you for acknowledging the implementation of some of your previous comments and apologize if some were not addressed adequately. We read your round two review comments carefully and did our best to address them as much as possible.
- One of the significant limitations of this pilot study with a small internal grant funding is that the control group’s sample size was smaller than the patient group. As mentioned before, in this study, we did not plan to recruit any subjects, and it was entirely dependent on another study with its specific protocol for subject sample collection. The collected plasma samples were used for other experiments in the original research; in some cases, there was not enough sample left to be used in the proposed analysis in the current study. This was more of the case for healthy control samples.
We plan to address such limitations by designing more extensive research and enrolling a statistically sound number of subjects to confirm our findings.
- In compliance with your comments on the incorrect interpretation of insignificant differences, we removed those inaccurate statements and rewritten sections 3.5 and 3.6 as below:
3.5 Effect of biological variables on the RAS Components and ArA metabolites level
Based on the age of the patients, they were divided into geriatric (≥ 65) and non-geriatric (< 65). Ang 1-7, Ang II levels and their ratio were not affected by the age of the patients (Figure 4, upper panel). Similarly, the Ang 1-7, Ang II levels and the ratio of Ang 1-7/Ang in male patients were comparable to female patients. Body weight differences in COVID-19 patients did not impact the Ang 1-7 and Ang II levels. A decrease in Ang 1-7/ Ang II ratio in patients with BMI > 30 was insignificant (Figure 4, lower panel).
3.6 Effect of comorbidities on the RAS and ArA pathway
The impact of existing comorbidities (respiratory, cardiovascular, hypertension or diabetes ) on the RAS components and ArA metabolites was tested in COVID-19 patients. Our data indicate no significant differences in the level of the RAS components in patients with and without comorbidity (Figure 5). Similarly, in the case of ArA metabolites, the effect of the presence or absence of comorbidities was insignificant (data not shown).
- Thank you for your valuable comment and suggestion regarding cross-referencing literature reports about the levels of Ang 1-7 and Ang II in COVID-19 patients and controls. We apologize for missing such an important point while there is some controversy around this topic. We included an extra seven references representing both sides of the debate and added some possible explanations and justification of the reported results in different studies. Such information was included in the discussion section below:
Our results indicate that Ang 1-7 levels were significantly lower, whereas Ang II levels were prominently higher in the COVID-19 patients than in the healthy control individuals (Fig. 3 and Table 2). The ratio of Ang 1-7/Ang II as an indicator of the balance between the RAS classical and protective arms was dramatically lower in COVID-19 patients. Such imbalance is attributed to the deactivation of the ACE2 enzyme by the SARS-CoV-2 virus and matched with measured higher Ang II plasma levels in these patients. Our study’s results agree with most of the previous reports that showed that the protective arm of the RAS is downregulated due to the infection [36-39]. In concert with our findings, these studies found significantly lower circulating Ang 1-7 levels in COVID-19 patients than in the control group. They also reported that the Ang II levels were lower in the patients that recovered from the disease than those who succumbed to the infection, and similarly, concluded that the elevated ratio of Ang II/Ang 1-7 is an indication that the balance of the RAS shifted towards the classical axis with increasing hospitalization, disease severity and mortality [36]. In another study, Lieu et al. found plasma Ang II levels to be elevated in the COVID-19-infected groups compared to healthy controls [38]. Additionally, the study of Wu et al. determined that the circulating plasma concentration of Ang II increased with the severity of the disease [39].
Although our study results align with most recently reported literature [36-39], few studies have reported that Ang 1-7 levels were elevated in the infected group rather than in the controls [40-42]. The observed controversy could be explained by different methodological approaches in patient enrolment or sample collection and analysis, among other possible explanations. Martins et al. analyzed plasma samples from individuals already taking ACEi or ARBs [40]. These classes of drugs are known to increase ACE2 activity, elevate Ang 1-7 and reduce Ang II plasma levels. Such an effect is expected to be more pronounced in healthy controls as no virus is present to inhibit ACE2 enzyme activity, somehow explaining the Martins et al. observations [40]. A comprehensive meta-analysis showed that the levels of Ang 1-7 in some studies were approximately 11-fold higher in the infected group than in the control group. It also included other controversial studies that reported the other way around. Interestingly, this meta-analysis founds that the level of Ang II was also elevated in the COVID-19-infected group[41]. The observed contradiction could have occurred due to including a range of studies that utilized different sample collection and processing methods. Some of those studies did not consider the enzymes and peptide stability, and some used nonspecific analytical methods, as highlighted by Martins et al. [40]. In another small study (enrolling 10 patients and 5 healthy control subjects), authors looked at equilibrium analysis reflecting overall plasmatic RAS activity instead of its actual status as they did not consider adding protease inhibitors to plasma samples [42]. That means the reported results is not representing the actual status of the RAS components at the time of sample collection. Therefore, it is not a direct measure of RAS peptides but measures the ongoing activity of RAS proteases which could make a significant difference in the level of peptides. It is worth mentioning that the RAS is a complex system comprised of non-ACE pathways that could impact the Ang II and Ang1-7 levels when ACE2 activity was influenced by COVID-19 infection [41].
Thank you again for taking time out of your busy schedule to read this manuscript’s original and revised versions and our responses to your constructive comments. We hope our response covers your raised issues and that you see them as convincing.
Best Regards,

Reviewer 2 Report
The authors have sufficiently addressed the comments of the reviewer and have revised the manuscript accordingly.
Author Response
We are grateful for the reviewer’s constructive comments and guidance in improving our manuscript’s quality.
Reviewer 3 Report
The paper has been extensively revised and can now be accepted for publication.
Author Response

(The authors gave the same response as above.)
